# A Comprehensive Review of Factors That Influence the Accuracy of Intraoral Scanners

**DOI:** 10.3390/diagnostics13213291

**Published:** 2023-10-24

**Authors:** Lubna Alkadi

**Affiliations:** 1Restorative and Prosthetic Dental Sciences, College of Dentistry, King Saud bin Abdulaziz University for Health Sciences, National Guard Health Affairs, Riyadh 11426, Saudi Arabia; lubna.alkadi@gmail.com; 2King Abdullah International Medical Research Center, National Guard Health Affairs, Riyadh 11481, Saudi Arabia

**Keywords:** accuracy, digital impressions, digital scans, intraoral scanners, precision of intraoral scanners, trueness of intraoral scanners

## Abstract

Intraoral scanners (IOSs) have become increasingly popular in the field of dentistry for capturing accurate digital impressions of patients’ teeth and oral structures. This study investigates the various factors influencing their accuracy. An extensive search of scholarly literature was carried out via PubMed, utilizing appropriate keywords. Factors evaluated in the included studies were categorized into three primary divisions: those related to the operator, the patient, and the IOS itself. The analysis demonstrated that the accuracy of intraoral scanning is influenced by various factors such as scanner selection, operator skill, calibration, patient’s oral anatomy, ambient conditions, and scanning aids. Maintaining updated software and understanding factors beyond scanner resolution are crucial for optimal accuracy. Conversely, smaller IOS tips, fast scanning speeds, and specific scanning patterns compromise the accuracy and precision. By understanding these factors, dental professionals can make more informed decisions and enhance the accuracy of IOSs, leading to improved final dental restorations.

## 1. Introduction

The most significant evolution in dentistry in recent years is manifested in the emergence of digital dentistry [1]. The seamless integration of advanced digital technologies has revolutionized the landscape of this field, with intraoral scanners (IOSs) emerging as pivotal game-changers. The development of IOSs was driven by the aspirations to improve traditional impression-taking processes, which are frequently subject to human error, with an ultimate goal of rendering them less technique sensitive [2].

The introduction of IOSs has signaled a paradigm shift in the field, offering substantial advantages that span from increased patient comfort to the production of highly accurate dental restorations [1,3]. As a result, these digital technologies have earned profound recognition in the profession, serving as dependable, precision-oriented instruments for recording dental structures.

The accuracy of IOSs is not merely a desirable trait, rather, it is an absolute requisite [2,4]. The precision of these devices critically influences the integrity and fit of final dental restorations, directly impacting their functional performance and esthetic outcome [2,4]. Inaccurate scans can lead to restoration misfit, which may compromise periodontal health, function, and longevity of the restoration, further necessitating revisions and additional clinical sessions [5]. Therefore, assuring IOSs’ accuracy should remain paramount to the successful employment of these devices in clinical and laboratory settings.

The accuracy of IOSs has been characterized in the literature by two independent factors: trueness and precision. Trueness refers to the degree of variation between the shape captured by the tested impression method and the original geometry (as in comparing a reference master cast with the digitized model). In contrast, precision measures the extent of variations or deviations among impressions within a specific test group (as in intragroup comparison of digitized models) [5,6,7].

However, ensuring the accuracy of intraoral scanning is a multi-faceted task, influenced by a variety of factors. These encompass operator-related aspects, such as their skill, experience, and continued training [8]. Furthermore, patient-oriented variables like their cooperation, oral condition, and the nature and location of materials and preparations to be scanned can significantly affect the scanning accuracy [3,6,9,10,11,12,13]. The scanning strategy and environment, inclusive of lighting conditions [1] and chosen scanning protocol [2,14,15,16], also weighs in on the final scanning accuracy. Notably, elements integral to the scanner systems themselves [12,14,17], such as hardware capabilities, software versions [18], and scanning resolution [11], contribute to the overall efficacy of an IOS.

This review serves to elaborate on these different aspects, aiming for a holistic and comprehensive understanding of factors that influence IOS accuracy and outlining opportunities for its continuous optimization.

## 2. Materials and Methods

A comprehensive review of the literature was done to investigate the linkage between diverse factors and the accuracy of IOSs. The methodologies employed in these studies exhibited substantial divergence, rendering a meta-analysis unfeasible. Therefore, a scoping review was conducted. To ensure the thoroughness and relevance of the review, an extensive search of scholarly literature was carried out via PubMed, utilizing appropriate keywords. PubMed was selected as the main database for this search process due to its comprehensive and extensive coverage of biomedical literature, which is primarily maintained by the National Institutes of Health’s National Library of Medicine. The choice was driven by PubMed’s reputation for hosting high-quality, peer-reviewed articles, its rigorous resource authority, as well as its advanced search options. Its regular updates ensure that the latest studies are accessible, which enriches the timely relevance of this review. Furthermore, the open access nature of PubMed removes economic barriers to access. This search targeted the extraction of research articles pertaining specifically to factors influencing the accuracy of IOS. The following search query was utilized: ((((Accuracy) OR (precision)) OR (trueness)) AND ((Intraoral scan*) OR (scan*) OR (three-dimensional) OR (digital impression) OR (CAD/CAM))) AND (((((((oral) OR (dental)) OR (dent*)) OR (*oral)) OR (tooth)) OR (implant)) OR (abutment) AND (y_5[Filter])).

A study was considered eligible for inclusion in this analysis if it satisfied the following criteria: (a) A rigorous evaluation of the relationship between an array of factors and the accuracy of IOSs; (b) The authorship and publication in English for comprehensive and clear interpretation, mitigating potential misinterpretations due to translation; (c) The research having been executed within the past five years (2018–2023), ensuring the incorporation of the most contemporary and relevant insights; (d) A strict concentration on IOSs, rather than a generalized focus on dental scanning technology; (e) Providing unambiguous, quantifiable outcomes related to the impact of examined factors on the accuracy of IOSs. Conversely, exclusion of studies was based on (a) their status as case reports, which frequently encapsulate singular, unique instances as opposed to generalizable findings; (b) the absence of a clear focus on factors impacting IOS’s accuracy; (c) the lack of adequate methodological detail, which may have undermined the reliability and credibility; (d) their categorization as opinion pieces, ensuring the analysis was rooted in primary, original research.

## 3. Factors Influencing the Accuracy of IOS

### 3.1. Operator-Related Factors

When exploring IOSs’ accuracy, one must acknowledge various factors significantly influenced by operator decisions. These include the operator’s proficiency with the device [19], operating distance and angulation [20,21,22], the scanning pattern or sequence [14,15,23,24], and protocols used for establishing cut-off points, re-scanning, and overlap [25,26,27,28,29]. A comprehensive understanding of these crucial influences is central to optimizing the scanning accuracy of IOSs and successfully incorporating these devices into regular dental practice. Dental professionals should appreciate and comprehend these factors fully to maximize the utilization of IOSs [8].

#### 3.1.1. Operator’s Proficiency with the IOS

Several studies have demonstrated the impact of operator experience on the accuracy of scans generated by different IOSs. Lim et al. [19] conducted a study to evaluate the impact of dental practitioner’s ‘experience curve’ on the precision and trueness of full-arch scans using two different IOSs, TRIOS (3Shape, Copenhagen, Denmark), and iTero (Align Technology, San Jose, CA, USA). Twenty dental hygienists, grouped based on their years of experience, scanned the assigned patients’ oral cavity ten times each. The findings showed that the Trios scanner outperformed the iTero scanner in terms of precision. Yet, the iTero scanner improved in trueness with repeated use, a trend not observed with the Trios. Interestingly, longer clinical experience significantly enhanced the trueness of the iTero scans after multiple uses, especially in the maxillary arch [19]. Similarly, in a study by Revell et al. [30] the performance of various IOSs was evaluated during full-arch implant scanning. The results showed that the accuracy of the scans differed among the scanners, and the experience of the operator played a significant role in reducing errors. The deviation from the implant platform was higher when a less experienced operator performed the scans compared to experienced operators [30]. In a study done by Resende et al. [31] comparing the performance of two IOSs, CEREC Omnicam (Dentsply Sirona, Charlotte, NC, USA) and TRIOS 3 (3Shape), three groups of operators with different levels of experience scanned a master model multiple times. The precision and trueness of the scans were evaluated by comparing them with reference scans. The results showed that operator experience, scanner type, and scan size all influenced the accuracy of the scans. More experienced operators achieved faster and more accurate scans, and smaller scan sizes also contributed to higher accuracy. Overall, the study found that the TRIOS 3 (3Shape) scanner provided more accurate complete-arch scans compared to the CEREC Omnicam (Dentsply Sirona) [31]. Taken together, the studies highlight the importance of both the choice of scanner and the expertise of the operator in achieving accurate implant scanning results.

#### 3.1.2. Scanning Distance and Angulation

Based on the available evidence, it is clear that various scanning parameters, particularly scanning distance and angulation, impact the accuracy of IOSs. The study conducted by Rotar et al. [20] using i700 (Medit, Seoul, Republic of Korea) IOS, showed that a scanning distance of 10 mm presented the highest accuracy; however, they also highlighted that precision decreases as the scanning distance increases [20]. This aligns with findings from Kim et al.’s [21] earlier study using TRIOS 3 (3Shap), CS 3500 (Carestream, Rochester, NY, USA), and PlanScan (Planmeca, Helsinki, Finland), which indicated peak accuracy at scanning distances of 2.5 mm and 5.0 mm, while a distance of 0 mm was the least accurate [21]. Furthermore, the impact of scanning angulation was highlighted in Button et al.’s [22] study using four different IOSs, i700 (Medit), TRIOS 4 (3Shape), CS 3800 (Carestream), and iTero (Align Technologies). The study demonstrated improved accuracy and a greater scanning area at a 0-mm scanning distance combined with a 15-degree scanning angulation. Overall, these findings indicate the need for calibrated scanning parameters that optimize the accuracy of individual IOSs [22].

#### 3.1.3. Scanning Sequence

Scanning sequence has an influence on the trueness and precision of IOSs. Studies by Diker et al. [14,23] have revealed the impact of scanning sequence on the trueness and precision of the scan, particularly when scanners like iTero (Align Technology), Virtuo Vivo (Dental Wings, Montreal, QC, Canada), Planmeca Emerald (Planmeca), and Primescan (Dentsply Sirona) are in question. Similarly, Oh et al. [15] reported that different scanning strategies, such as vertical rotation, continuous horizontal, and segmental, can alter scan trueness significantly, though the precision seemed unaffected and scanner type did not appear to influence the outcomes [15]. Meanwhile, research by Passos et al. [24] showed that while certain scanning strategies tended to perform best with specific scanners, as with the Linear-continuous group or the M group with the Primescan (Dentsply Sirona), others, like the CEREC Omnicam (Dentsply Sirona), did not exhibit a singular superior strategy for trueness and precision [24]. Overall, these studies emphasize the importance of careful selection of scanning sequence to ensure optimum outcomes. Generally, it is advisable to adhere to the manufacturer’s guidelines on the scanning strategy for individual IOSs.

#### 3.1.4. Cut Out-Rescan and Overlapping Procedures

Rescanning, as a method to acquire data from previously unscanned regions, completes the 3D mesh geometry of scanned surfaces. In particular, it is often recommended in the digital manufacturing workflow for tooth- and implant-supported prostheses to fill scan gaps or ‘mesh holes’. However, the complexities of the rescanning process, such as cutting off and re-acquiring data, or alternatively overlapping new data over pre-existing scans, have shown varied implications on scanning accuracy [25,26]. Research conducted by Revilla-León et al. [26,27] demonstrated that alterations to the preexisting intraoral digital scan through rescanning procedures significantly decreased the overall scanning accuracy, with the number and diameter of rescanned areas playing a crucial role. Gómez-Polo’s [28] findings reinforced these findings, pointing out the negative effects of rescanning mesh holes and stitching procedures on both the trueness and precision of the IOSs tested. Interestingly, when observing different technologies, data exchange by over scanning with Primescan (Dentsply Sirona) has shown superior accuracy compared to the cut-out–rescan method; however, the latter produces more accurate results when used with CEREC Omnicam (Dentsply Sirona) [25]. Yet, Reich et al.’s [29] research introduced a contrasting perspective, suggesting that while scanners might differ in accuracy, the “cut out-rescan” procedure did not affect the accuracy within each IOS.

### 3.2. Patient-Related Factors

Patient factors refer to the specific intraoral conditions of the patient, which significantly impact the accuracy of IOSs [13]. Such influential factors include characteristics of the scanned area within the oral cavity [10,11,12,17,32,33], tooth preparation design [3,4,6,9,34,35,36,37], restorative materials, surface treatments, and wetness conditions of the digitized surfaces [7,38,39,40,41]. Implant-specific parameters such as interimplant distance, and the position, angulation of existing implants, and scanbodies also influence accuracy of the scans [13,42,43,44].

#### 3.2.1. Characteristics of the Scanned Area

##### Location of the Scanned Area

Numerous studies have evaluated how the accuracy of resulting scans is influenced by the location of the scanned area. An experimental study by Treesh et al. [10] evaluated the accuracy of four distinct IOSs, namely the CEREC Bluecam (Dentsply Sirona), CEREC Omnicam (Dentsply Sirona), TRIOS Color (3Shap), and CS 3500 (Carestream). They reported that the accuracy of scans is significantly influenced by the location of the scanned area. Benchmarked against a precisely 3D-printed reference cast, all IOSs consistently underestimated the dimensions of the reference file, affecting the trueness of the scans. This underestimation became more prominent within the posterior regions of the dental arch, where the local errors stretched beyond 100 μm for every scanner analyzed [10].

An in vivo study by Pellitteri et al. [12] investigated the accuracy of various intraoral scanning systems compared to the conventional polyvinyl siloxane (PVS) impression technique, focusing specifically on the location’s impact on scan trueness. Results demonstrated discrepancies in the range of 100–200 μm between digital and conventional impressions, with noticeable imprecision in the molar area of both dental arches. Although the CEREC (Dentsply Sirona) system tended to undersize the impression dimensions, the TRIOS (3Shape) showed superior single-tooth precision, and the CS 3600 (Carestream) outperformed in capturing inter-arch diameter. The inconsistencies were more evident in linear measurements, with a greater deviation in the molar region across all scanning methods. This affirmed the presence of distortion in digital impressions within the posterior region, with the TRIOS (3Shape) performing closest to the real dimension [12].

A study by Chiu et al. [11] explored the accuracy of digital dental impressions generated by TRIOS 3 (3Shape) used at different resolutions. Despite distinct variations in scan times and number of images recorded across high resolution, standard resolution, and combined resolution settings, no significant discrepancies were found on the preparation finish line for the all-ceramic crown of a mandibular molar across these resolutions. However, location-specific differences in accuracy were noted, with distal surfaces indicating the highest discrepancies—highlighting the influence of tooth surface location on the trueness of the scans. This signifies that, irrespective of the resolution setting, scan accuracy is notably impacted by the anatomical position of the tooth surface, an attribute that dental professionals should consider when employing IOSs [11].

##### Extension of the Scanned Area

A systematic review by Abduo et al. [32] based on 32 included studies examined the accuracy of various IOSs for dental impressions, analyzing factors influencing their performance. Full-arch scanning showed a potential for more deviations compared to partial-arch scanning [32].

Similarly, an in vitro study by Zimmerman et al. [17] evaluated the accuracy of several IOSs in obtaining full and partial-arch dental impressions. Traditional impressions using PVS were compared with eight different IOSs. Full-arch impressions proved challenging for IOSs with decreased trueness and precision values. However, partial-arch impressions, specifically for the posterior segment, demonstrated promising precision levels, matching conventional methods. These findings suggest that while specific IOSs can be a viable alternative to traditional methods for partial-arch impressions, full-arch impressions still present challenges for IOSs [17].

##### Arch Width

A study by Kaewbuasa et al. [33] compared the accuracy of three different IOSs: TRIOS 3 (3Shape), True Definition (3M ESPE), and Dental Wings (Dental Wings Inc.), considering varying dental arch widths. The accuracy, denoted in terms of trueness, differed significantly across these systems. Dental Wings (Dental Wings INC) registered significant relative length and angular deviations in smaller to medium-sized dental arches when compared with the TRIOS 3 (3Shape) and True Definition (3M ESPE) scanners. However, it showed enhanced accuracy when used with larger dental arches. On the other hand, True Definition has shown a tendency to cause inaccuracies when used on larger arches. TRIOS 3 (3Shape) consistently maintained its trueness across dental arches of all sizes, indicating its robustness towards size variations. This highlights that the accuracy of full-arch scans produced by certain IOSs may indeed be affected by the dimension of the dental arch being scanned. It is important to note that larger dental arches necessitate a greater scanning area than their smaller counterparts, which makes it impossible for an IOS to capture the entire arch in one go. Hence, multiple overlapping scans are performed and amalgamated utilizing stitching algorithms, which could introduce additional discrepancies [33].

#### 3.2.2. Tooth Preparation Design

Extensive research has emphasized the impact of preparation design on IOS accuracy, including aspects like the preparation type, finish line placement, tooth geometry, complexity, and finishing procedures [4,6,9,34,35,37].

Ashraf et al. [6] reported that for improved scan trueness, extracoronal preparations were favored over intracoronal ones, suggesting that preparation design plays a significant role. Additionally, subtle changes in tooth geometry such as increased convergence or divergence between opposing walls can improve trueness. The finding was explained by the preparations made with more straight, vertical walls being more prone to errors caused by camera misalignment, emphasizing the complex relationship between tooth geometry and scanning accuracy.

In terms of finish line placement, Son et al. [9] found significant variations in the accuracy of scans using the i500 (Medit) and EZIS PO (DDS) scanners, with equigingival and subgingival finish lines reporting poor accuracy. The study underlined that finish line location is a critical design feature affecting IOS’s accuracy. The accuracy was significantly improved with the use of gingival retraction cords. This is due to the cords creating a separation between the abutment and the gingiva by displacing the gingiva, effectively securing a sulcular width greater than 0.2 mm, which is essential for accurately scanning the prepared tooth [9].

The study conducted by Nedelcu et al. [34] compared the finish line distinctness (FLD) and finish line accuracy (FLA) across seven IOSs and traditional impressions. The study found TRIOS (3Shape) and CS 3600 (Carestream) to offer the highest FLD and FLA, respectively. Despite high accuracy from traditional impressions, both these and certain IOSs struggled in subgingival areas, highlighting the necessity of careful digital impressions evaluation due to varying technical limitations, especially in complex subgingival scenarios [34]. Abduo et al. [3] concluded that preparation types, whether inlay, onlay, or crown, significantly impacted the trueness of the scans. Among various preparations, inlays revealed the highest trueness, followed by crowns and onlays [3]. The complexity of the preparation design was another factor that was observed in the study by de Andrade et al. [35]. The accuracy of scans was influenced by the complexity of onlay preparation, where a simplified nonretentive design yielded higher trueness and precision than traditional, more complex designs [35]. Additionally, the study by Ammoun et al. [4] evaluated the accuracy of two different IOSs, TRIOS (3Shape), and True Definition (3M ESPE), in terms of distinct preparation designs and scan angle limitations due to adjacent teeth. The study demonstrated that partial coverage preparation scans were less accurate than complete coverage [4].

According to Revilla-León et al. [36], the nature of tooth preparation finishing also notably influenced the scanner’s accuracy. Preparation finishing procedures ranging from super-coarse grit, fine grit, air-particle abrasion, to immediate dentin sealing (IDS) exhibited differing IOS accuracies, with air-particle abrasion reflecting the highest and IDS the lowest scanning accuracy [36].

In contrast, Khaled et al. [37] indicated that different preparation depths for inlay-retained fixed dental prostheses did not significantly impact IOS accuracy. Regardless, the type of scanner used was an influential factor affecting accuracy, placing an additional focus on the importance of scanner selection, in addition to preparation design [37].

#### 3.2.3. Restorative Materials, Surface Treatments and Wetness Conditions

Revilla-León et al. [38] analyzed the impact of restorative dental materials and their specific surface treatments on the scanning accuracy of IOSs. Their findings uncovered considerable discrepancies in scanning accuracy across different materials. Restorative materials, such as conventional and milled Polymethyl Methacrylate (PMMA) and additively manufactured bis-acryl-based polymer, attained the highest scanning accuracy when they were polished. Conversely, high noble alloy specimens had the lowest trueness values. While glazed zirconia crowns had comparable trueness values to their polished counterparts, a polished surface in other materials resulted in better trueness than a glazed one. Precision was lowest in the conventional PMMA when it was glazed and highest in the polished bis-acryl composite resin. The findings suggest that both the type of restorative material and its surface treatment significantly influence the scanning accuracy of IOSs [38].

Yatmaz et al. [39] assessed the full-arch scan accuracy of four different IOSs, namely CEREC Omnicam (Dentsply Sirona), Primescan (Dentsply Sirona), TRIOS 4 (3Shape), and VivaScan (Ivoclar Vivadent, Schaan, Liechtenstein), on two distinct ceramic surfaces created from zirconium oxide and glazed lithium disilicate glass-ceramic. Notably, each IOS exhibited significant differences in trueness and precision, with Primescan (Dentsply Sirona) displaying the lowest deviation values in both categories for each material, while CEREC Omnicam (Dentsply Sirona) produced the largest values. Furthermore, CEREC Omnicam (Dentsply Sirona) and VivaScan (Ivoclar Vivadent) showed significantly different precision values depending on the scanned surface [39].

The study by Agustín-Panadero et al. [40] extended this exploration by investigating the impact of different restorative materials and varying levels of surface wetness on the accuracy of IOSs. The research used four groups identified by the first molar’s material: natural tooth, zirconia, lithium disilicate, and nanoceramic resin crown, further subdivided into dry, low-, mild-, and high-wetness categories. The TRIOS 3 (3Shape) IOS was utilized for all scans. Findings indicated that both the restorative material and surface wetness significantly affected scanning trueness and precision. Greater wetness levels resulted in lower trueness and precision, with dry and low wetness subgroups outperforming mild and high wetness counterparts. In terms of material impact, natural tooth, zirconia, and lithium disilicate exhibited superior trueness under dry and low wetness conditions compared to the nanoceramic resin crown group, although no significant precision difference was found across all materials. Under high wetness conditions, lithium disilicate demonstrated superior trueness and precision. These results suggest that drier surfaces are recommended to enhance scanning accuracy, and both the presence of saliva and the type of dental restorations may reduce the IOS’s performance [40].

Similarly, a study by Chen et al. [7] evaluated the impact of tooth surface wetness on the accuracy of IOSs and the effectiveness of drying methods. The study used a mandibular jaw model scanned under three conditions (dry, wet, and blow-dry), with either ultra-pure water or artificial saliva. They reported that wet conditions significantly impaired scanning accuracy in terms of both trueness and precision. Two IOSs were used, but the type of liquid did not significantly alter results. Scanning inaccuracies occurred primarily in the pits and fissures of the occlusal surface of posterior teeth, in interproximal areas, and the margin of abutments. Blow-drying the tooth surface before scanning effectively reduced these errors, which implies that the presence of liquid can affect IOS accuracy [7].

A study by Rapone et al. [41] also evaluated the accuracy of three commercially available IOSs in a wet oral environment model, using an in-vitro experiment of four permanent teeth with a total of 240 digital impressions. Results highlighted the influence of oral biological fluids on the precision of the digital impressions and the challenge of obtaining accurate scans under wet conditions [41].

Moreover, Michelinakis et al. [45] evaluated the accuracy of three distinct IOSs, namely TRIOS 3 (3Shape), CS 3600 (Carestream), and Emerald S (Planmeca), across an array of diverse dental materials. The results indicate the type of substrate significantly alters the performance of the TRIOS 3 (3Shape) and Emerald S scanners, especially when scanning more translucent and reflective materials. Among the tested scanners, the TRIOS 3 (3Shape) demonstrated greater trueness and precision in full-arch scanning compared to the CS 3600 (Carestream) and Emerald S (Planmeca); however, there was no significant difference between the latter two in terms of accuracy. The accuracy of all tested scanners was affected by the dental material substrates. Full-metal crowns were observed to have the lowest trueness in scanning across all devices. For high-translucency substrates, the TRIOS 3 (3Shape) displayed superior trueness relative to the CS 3600 (Carestream). The surface finish of Class II amalgam restorations did not noticeably impact trueness for any of the scanners. In terms of full-arch accuracy, the TRIOS 3 (3Shape) scanner outperformed the other two. Nevertheless, all scanners showed a mean full-arch accuracy below the 100 μm threshold [45].

In sum, these findings suggest that a careful understanding of the interplay between IOSs, restorative materials, and the oral environment is vital for optimizing scanning accuracy. Thus, decisions concerning the choice of scanner, treatment materials, and surface wetness conditions can significantly impact the outcome.

#### 3.2.4. Implants and Scanbodies

A study by Thanasrisuebwong et al. [42] assessed the impact of inter-implant distances on the accuracy of IOSs. Three models with different distances between two scan bodies were fabricated and then scanned using two IOSs. The results showed that inter-implant distance had a significant effect on both scanners. Trueness and precision decreased with longer inter-implant distances, but the distortions were not clinically significant [42]. A study by Gómez-Polo [43] examined how scan body geometry, bevel location, implant angulation, and position affect the accuracy of full-arch implant scans. Two definitive casts with implant analogs were used, and various subgroups based on bevel position were created. The results showed that bevel position and inter-implant distance affected linear discrepancies. Lingual orientation and parallel implant analog positions yielded better accuracy. Implants positioned where scanning finished had higher distortion compared to contralateral implants [43]. A study by Ashraf et al. [44] evaluated the accuracy of IOSs for scanning implant-supported full-arch fixed prosthesis with various implant angulations, both with and without splinting the scanbodies. Two maxillary models were used and divided into two groups based on the posterior implant angulation. Each group was further divided into subgroups based on the type of IOS used. Trueness and precision were analyzed, revealing that the angulation of the implants did not significantly affect accuracy, while splinting the scanbodies had a significant positive impact. The type of scanner also significantly influenced trueness and precision, with CEREC Primescan (Densply Sirona) demonstrating higher accuracy than TRIOS 4 (3shape) and Medit i600 (Medit) [44].

### 3.3. IOS-Related Factors

Available literature discuss various IOS-related factors that can impact the accuracy of scans. It provides insights into the influence of ambient lighting and temperature on scanning accuracy [46,47,48,49,50], as well as the importance of surface pre-treatment [51] and scan resolution [11,52]. The software versions and updates of the scanners are also highlighted [18,53], along with the impact of scanner head size on trueness and precision [54,55]. These findings emphasize the need to optimize scanning conditions and consider multiple factors when aiming for optimal accuracy in dental scanning processes.

#### 3.3.1. Scanning Environment

##### Ambient Lighting

Ambient lighting conditions can influence the accuracy (precision and trueness) of IOSs, but the ideal conditions vary depending on the selected IOS. In a study conducted by Revilla-Leon et al. [46], they reported that different lighting conditions yielded better results for different scanners. For the iTero Element (Align Technology), improved accuracy was observed under chair and room light conditions. For the CEREC Omnicam (Densply Sirona), optimal accuracy was attained under zero light conditions. Meanwhile, the TRIOS 3 (3shape) scanner exhibited superior accuracy under standard room light conditions [46]. Similarly, Ochoa-López et al. [47] conducted a study that evaluated the influence of ambient light conditions on the accuracy and scanning time of seven IOSs, particularly when used for full-arch implant scans. Their findings reveal that ambient lighting did indeed influence the accuracy and scanning time of IOSs assessed; however, the degree of influence was not uniform across all devices. Therefore, it is necessary to optimize ambient light illuminance for each specific IOS to boost the accuracy of scanning processes [47]. This was further echoed by Wesemann et al. [48] who also showed that ambient light impacts the accuracy and scanning timing of IOS. For 4-unit scans, the effect was not clinically significant. However, when it came to full-arch scans, they concluded that optimal lighting conditions could indeed enhance both the precision of scans and their completion time [48]. On the other hand, Jivanescu et al. [49] found that even though some variances were noticed in the trueness and precision data under different light intensities, these discrepancies were not clinically significant. Consequently, it was not plausible for them to conclude that ambient light has a substantial impact on the accuracy of intraoral scanning [49].

##### Ambient Temperature

Ambient temperature changes had a detrimental effect on the accuracy (trueness and precision) of the IOS tested in a study conducted by Revilla-leon et al. [50]. A full-arch maxillary dentate Type IV stone cast was digitized using an industrial scanner and a TRIOS 4 (3shape) IOS under different ambient temperature conditions and four groups were created based on temperature changes. The results showed that ambient temperature changes had a negative impact on the trueness and precision of the IOS. Increasing the ambient temperature had a greater influence on the scanning accuracy compared to decreasing the temperature. Thus, maintaining a stable ambient temperature is crucial for achieving accurate intraoral scans [50].

#### 3.3.2. Surface Pre-Treatment

A study conducted by Oh et al. [51] investigated the effect of scanning-aid materials on the accuracy and efficiency of full-arch scanning using IOSs. They used two types of scanning-aid materials for their experiments: IP Scan Spray (IP-Division, Haimhausen, Germany) and Vita Powder Scan Spray (Vita Zahnfabrik, Stuttgart, Germany). These were compared to a control group that had no treatment. The findings from the study showed that, statistically, there was a significant improvement in the precision of the scanned images when scanning-aid materials were used, compared to those from the no-treatment group. However, there was no difference observed between the trueness and the types of scanning-aid materials used. Another finding was related to the duration of the scanning process. They found that the use of scanning-aid agents significantly reduced the working time compared to the no-treatment group [51].

#### 3.3.3. Scan Resolution

A study conducted by Medina-Sotomayor et al. [52] provides insights into the resolution and accuracy of various IOSs used in digital dental impressions. According to their findings, CEREC Omnicam (Densply Sirona) demonstrated the highest resolution among the scanners tested, followed by True Definition (3M ESPE), TRIOS (3shape) and iTero (Align Technology). The study did not find a significant relationship between resolution and accuracy of the IOS, except for CEREC Omnicam (Densply Sirona) and its precision. This study concluded that resolution has no direct correlation with the accuracy of capturing a full-arch scan. It suggests that other factors beyond resolution alone, such as software algorithms, scanning technique, and material properties, may influence the overall accuracy of digital impressions [52]. According to research by Chiu et al. [11], the resolution of the scanner seems to be mostly determined by the hardware of the system and is optimized for default scans. They also suggest that utilizing a software high-resolution mode, which captures more data over a longer period of time, may not necessarily improve the scan accuracy [11].

#### 3.3.4. Software Versions and Updates

The impact of different software versions on the accuracy of the same IOS was studied by several research groups. By studying the CEREC Omnicam (Densply Sirona) with software versions 4.4.0 and 4.4.4, Haddadi et al. [53] reported that software version can significantly affect the accuracy of an IOS. They concluded that it is important for researchers to include the software version of scanners when publishing their findings. This information allows other researchers and practitioners to replicate the study accurately and understand the potential variations in results due to different software versions [53]. Similar findings were confirmed by Schmalzl et al. [18]. They conducted a study comparing different software versions of the TRIOS (3shape) IOS. They found that updating the software can have a significant positive impact on the trueness and precision of the IOS. They concluded that appropriate software updates can significantly increase the trueness and precision of IOSs. With updated software, the older generation can match the accuracy level of latest equipment. This highlights the importance of keeping the software up to date to ensure optimal performance [18].

#### 3.3.5. Scanner Head Size

In a study by Hayama et al. [54] comparing digital and conventional impressions for removable partial denture fabrication, researchers used mandibular Kennedy Class I and III models with soft silicone simulated-mucosa on edentulous ridges. Digital impressions obtained with IOSs had superior trueness but inferior precision compared to conventional impressions. The larger scanning head of the IOS showed better trueness and precision and required fewer scanned images. Overall, digital impressions showed promise but still had some areas for improvement when compared to conventional impressions for accuracy in removable partial denture fabrication [54]. Similarly, in a study by An et al. [55] 120 intraoral scans were conducted using different variables, including tip size, scanning pattern, and scanning speed. The scans were evaluated for trueness and precision using image analysis software. The findings showed that tip size significantly affected trueness, with smaller tips resulting in lower trueness. In terms of precision, all three variables had a significant influence, with smaller tips, faster scanning speeds, and an S-shaped scanning pattern leading to less precise scans. Overall, the use of a smaller tip negatively impacted both trueness and precision, while fast scanning speeds and an S-shaped scanning pattern resulted in lower precision compared to regular or slow scanning speeds and the occlusal-first scanning pattern [55].

## 4. Conclusions

Ensuring the accuracy of IOSs is a multifaceted task that involves various factors. These include operator-related, patient-related, and IOS-related variables. This review found that the accuracy of IOSs is significantly influenced by several critical factors, including the choice of scanner, operator’s expertise, calibration of scanning parameters, and unique aspects of the oral anatomy. Rescanning procedures can reduce the overall scanning accuracy, necessitating adherence to manufacturers’ guidelines. While IOSs may show promise in partial-arch impressions, complete-arch impressions remain challenging due to larger arch sizes needing multiple overlapping scans, potentially leading to additional discrepancies. Factors like ambient lighting and temperature also substantially affect accuracy and scanning time, although not uniformly across devices. The utilization of scanning-aid agents can reduce working time; however, their effect on trueness was not consistently significant. The results also imply that higher resolution does not guarantee more accurate full-arch scans, suggesting the need for a comprehensive understanding of parameters like software algorithms, scanning techniques, and material properties. Importantly, keeping the software updated can help achieve optimal performance. Contrarily, smaller tips, fast scanning speeds, and certain scanning patterns can negatively influence accuracy and precision. By understanding and addressing these factors, dental professionals can enhance the accuracy of IOSs, leading to improved clinical outcomes, reduced restoration misfit, and increased patient satisfaction. Further research and advancements in this field are warranted to continually refine the accuracy and efficacy of intraoral scanning technology.

## Data Availability

Not applicable.

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
