# Peer review of "A Comprehensive Review of Factors That Influence the Accuracy of Intraoral Scanners"

_diagnostics, 2023, doi:10.3390/diagnostics13213291_

Round 1

Reviewer 1 Report

The manuscript entitled: “A Comprehensive Review of Factors that Influence the Accu-
racy of Intraoral Scanners” presents an insight into the relations between the intraoral scan and clinical success.  This topic is highly relevant and article may be very helpful for dentists. However there are some questions that I would like to ask.

I would like to know what key ward were used while searching PubMed?

Why Pub Med was the chosen data base?

I would be beneficial for the reader to add to the Conclusions some advises for the dental professionals for better accuracy of the intraoral scans.

Author Response

Dear Respected Reviewer,

Thank you for the effort and time spent reviewing my manuscript. Please find my response to each of your kind comments attached as a PDF.

Sincerely,

Reviewer 2 Report

Dear Authors,

Congratulations on the work you have done and presented in this manuscript. I believe that your work is of high quality and interest for the general reader. This is a well written, comprehensive review, and therefore I will recommend publication of your article after some minor corrections. 

First and foremost: there are multiple grammar and spelling errors, English needs to be revised. Secondly, there are some minor issues regarding references citation in text, small errors, big impact. Please correct it.  Please see the attachment. 

Moderate changes are required

Author Response

(The authors gave the same response as above.)
